

# Natural selection drives chemical resistance of *Datura stramonium*

Adán Miranda-Pérez[1], Guillermo Castillo[1,2], Johnattan Hernández-Cumplido[3], Pedro L. Valverde[4], María Borbolla[1], Laura L. Cruz[1], Rosalinda Tapia-López[1], Juan Fornoni[1], César M. Flores-Ortiz[5] and Juan Núñez-Farfán[1]

[1] Laboratory of Ecological Genetics and Evolution, Department of Evolutionary Ecology, Institute for Ecology, Universidad Nacional Autónoma de México, Mexico City, México
[2] Laboratorio Nacional de Análisis y Síntesis Ecológica, Escuela Nacional de Estudios Superiores, Universidad Nacional Autónoma de México, Campus Morelia, Michoacán, México
[3] Department of Entomology, Rutgers University, P.E. Marucci Center, Chatsworth, NJ, United States
[4] Department of Biology, Universidad Autónoma Metropolitana, Mexico City, México
[5] UBIPRO, Facultad de Estudios Superiores Iztacala Universidad, Nacional Autónoma de México, Tlalnepantla, México

Corresponding author
Juan Núñez-Farfán, farfan@unam.mx

## ABSTRACT

Plant resistance to herbivores involves physical and chemical plant traits that prevent or diminish damage by herbivores, and hence may promote coevolutionary arm-races between interacting species. Although *Datura stramonium's* concentration of tropane alkaloids is under selection by leaf beetles, it is not known whether chemical defense reduces seed predation by the specialist weevil, *Trichobaris soror*, and if it is evolving by natural selection. We measured infestation by *T. soror* as well as the concentration of the plants' two main tropane alkaloids in 278 *D. stramonium* plants belonging to 31 populations in central Mexico. We assessed whether the seed predator exerted preferences on the levels of both alkaloids and whether they affect plant fitness. Results show great variation across populations in the concentration of scopolamine and atropine in both leaves and seeds of plants of *D. stramonium*, as well as in the intensity of infestation and the proportion of infested fruits by *T. soror*. The concentration of scopolamine in seeds and leaves are negatively associated across populations. We found that scopolamine concentration increases plant fitness. Our major finding was the detection of a positive relationship between the population average concentrations of scopolamine with the selection differentials of scopolamine. Such spatial variation in the direction and intensity of selection on scopolamine may represent a coevolutionary selective mosaic. Our results support the view that variation in the concentration of scopolamine among-populations of *D. stramonium* in central Mexico is being driven, in part, by selection exerted by *T. soror*, pointing an adaptive role of tropane alkaloids in this plant species.

## INTRODUCTION

The coevolutionary process involves reciprocal selection-adaptation between interacting species through time (*Dawkins & Krebs, 1979*; *Thompson, 1999*; *Thompson, 2005*). This

adaptation and counter-adaptation phenomenon could result in a coevolutionary arms race, a process of offense-defense (*Dawkins & Krebs, 1979*). A coevolutionary relationship between plants and herbivores may generate symmetrical or asymmetrical selective pressures between interacting species. These selective pressures may be different spatially and could produce a geographic structure of interactions (*Forde, Thompson & Bohannan, 2004*; *Gomulkiewicz et al., 2002*; *Nuismer, Thompson & Gomulkiewicz, 2000*; *Thompson, 1999*; *Thompson, 2005*). In some locations the interacting species exert reciprocal selection pressures to one another (coevolutionary hot spots), whereas in other locations reciprocal selection is highly asymmetric (coevolutionary cold spots) (*Gomulkiewicz et al., 2002*; *Nuismer, Thompson & Gomulkiewicz, 2000*; *Thompson, 1999*; *Thompson, 2005*).

A coevolutionary arms race between herbivores and plants may be favored in specialized interactions as in the case of insects that detoxify specific compounds (*Janzen, 1969*; *Janzen, 1973*; *Schoonhoven, Van Loon & Dicke, 2005*). For instance, the aphid *Macrosiphum albifrons* is adapted to consume *Lupinus angustifolius* with a low content of alkaloids (*Philippi et al., 2015*); however, dietary specialist herbivore insects may also be adapted to tolerate secondary metabolites of their host plants without switching to other different host plants (*Laukkanen et al., 2012*). In *Arabidopsis thaliana*, the abundance of two aphid herbivore species is correlated to a genetic polymorphism of the plant's resistance locus. This polymorphism is under selection due to changes in population size of the two aphid species (*Züst et al., 2012*).

Host-parasite systems, that exert reciprocal selection pressures, offer the opportunity to assess the asymmetry in selection pressures and the potential to produce adaptation (*Greischar & Koskella, 2007*). Local adaptation may produce population differentiation as a by-product of natural selection (*Kawecki & Ebert, 2004*). This process is well illustrated by the weevil *Curculio camelliae* (Coleoptera: Curculionidae) that parasitizes the fruits of *Camellia japonica* (Theaceae) (*Iseki, Sasaki & Toju, 2011*; *Toju, 2007*; *Toju, 2009*; *Toju & Sota, 2006*). The fruits of *C. japonica* are capsules with a thick pericarp, dehiscent, with three locules and one seed per cavity (*Okamoto, 1988*). Females of *C. camelliae* perforate the thick pericarp with its long *rostrum*, modified labial cavity in insects (*Resh & Cardé, 2009*), and oviposit on the seeds (*Toju, 2007*). A successful weevil infestation, or oviposition, depends on the phenotypic match between the rostrum length and pericarp thickness. These two traits that mediate the interaction vary geographically and are under selection (*Toju, 2007*; *Toju, 2009*; *Toju & Sota, 2006*). Some evidence, however, indicates that these phenotypic characteristics may also vary according to abiotic factors, i.e., the latitude (*Iseki, Sasaki & Toju, 2011*). Furthermore, infestation by the weevil *C. camelliae* increases at higher-altitude localities and its obligated host plant decreases its resistance (*Toju, 2009*). In the *C. camelliae*-*C. japonica* system, natural selection acts on pericarp thickness that is a physical barrier that prevents infestation by weevils (*Toju, 2007*; *Toju, 2009*; *Toju & Sota, 2006*).

In the annual herb *Datura stramonium*, tropane alkaloids function either as resistance characters preventing foliar damage by herbivores and/or as phagostimulants to them (*Castillo et al., 2013*; *Castillo et al., 2014*; *Shonle & Bergelson, 2000*). Evidence shows that alkaloid concentration in *D. stramonium* varies across populations (*Castillo et al., 2014*)

and that such differentiation in chemical defense could be adaptive (*Castillo et al., 2015*). In some populations, dietary specialist and generalist folivores select against atropine concentration, whereas scopolamine is positively selected selected by the dietary specialist folivore *Lema daturaphila* and by the generalist grasshopper *Sphenarium purpurascens* (*Castillo et al., 2014*). Fruits of *D. stramonium* are parasitized by *Trichobaris soror* (Coleoptera: Curculionidae) that reduces plant fitness by consuming the seeds (*Cabrales-Vargas, 1991*; *Cruz, 2009*; *De-la-Mora, Piñero & Núñez-Farfán, 2015*). However, to what extent alkaloids of *D. stramonium* could affect infestation by the seed predator is not known. Here, we analyzed the relationship between tropane alkaloids produced by *D. stramonium* and infestation by the specialized seed predator *T. soror* across multiple populations in central Mexico. We aimed to determine whether *D. stramonium*'s tropane alkaloids prevent infestation by *T. soror*. Specifically, we addressed the following questions: 1. Are alkaloids resistance characters that prevent/reduce infestation by weevils; 2. Do seed predators exert natural selection upon plant's alkaloids concentration; and 3. Do variation of both alkaloid concentration and infestation by weevils across populations is correlated to the localities' environmental conditions? (*v. gr.*, *Toju, 2009*).

## MATERIALS AND METHODS

### Study system

The weevil *Trichobaris soror* (Coleoptera: Curculionidae) is intimately associated to the life cycle of *D. stramonium* (*Bello-Bedoy, Cruz & Núñez-Farfán, 2011a*; *Borbolla, 2015*; *Cabrales-Vargas, 1991*). *Trichobaris soror* is distributed mainly in central Mexico (*Barber, 1935*; *De-la-Mora, Piñero & Núñez-Farfán, 2015*); adult weevils feed on leaves, calyx and floral tissues of *D. stramonium*. After mating, females oviposit at the base of developing fruits. Their larvae feed exclusively on immature seeds inside the developing fruit where they build tunnels with their own feces. Larvae pupate in the fruit and sometimes are parasitized by wasps. The weevils hibernate inside the fruit of *D. stramonium* until the next rainy season (*Bello-Bedoy, Cruz & Núñez-Farfán, 2011a*; *Borbolla, 2015*; *Cabrales-Vargas, 1991*).

Besides the seed predator, *D. stramonium* (Solanaceae) is preyed upon by specialist leaf-beetles (*Lema trilineata* and *Epitrix parvula*, Coleoptera: Chrysomelidae). This weed species has been widely studied in relation to its resistance characters (alkaloids, leaf trichomes) against these leaf herbivores (*Bello-Bedoy & Núñez-Farfán, 2011b*; *Cabrales-Vargas, 1991*; *Carmona & Fornoni, 2013*; *Castillo et al., 2013*; *Castillo et al., 2014*; *Núñez-Farfán & Dirzo, 1994*; *Shonle & Bergelson, 2000*; *Valverde, Fornoni & Núñez-Farfán, 2001*). However, it is unknown if chemical defense of *D. stramonium* prevents the infestation by the weevil *T. soror*.

### Sampled populations

During the reproductive season of *D. stramonium* (September-November) in 2007, we sampled different populations across Central Mexico. We collected an average of 30 plants from 31 populations (Fig. 1, Table S1). For each plant, all mature fruits were collected and individually bagged and labeled. Before opening, the width and length of each fruit was

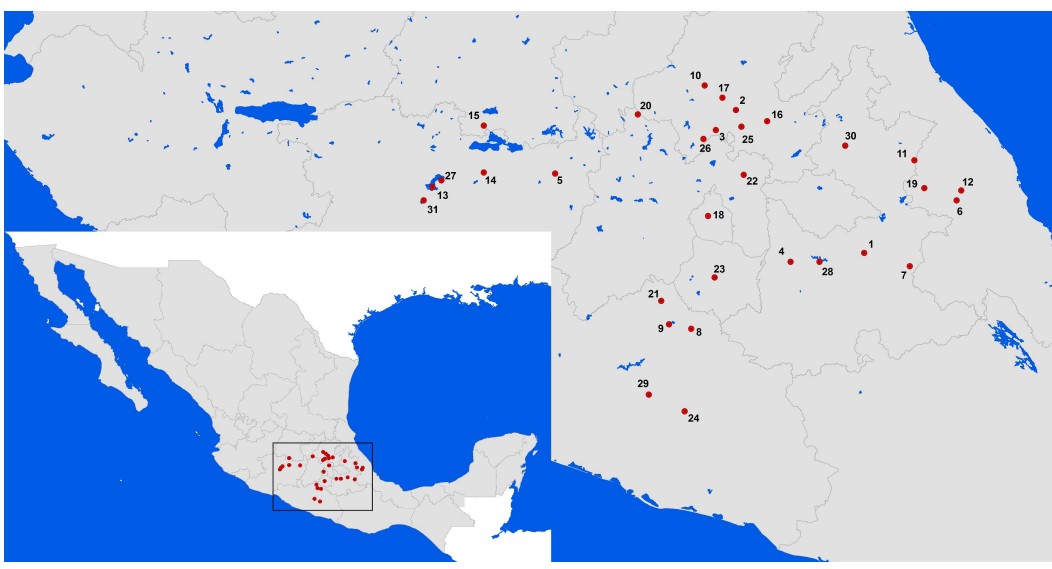

**Figure 1** *Datura stramonium* **populations sampled in Central Mexico.** The number each population corresponds to the locality given in Table S1.

measured to calculate its volume with the equation:

$$V = \left(\frac{4}{3}\right)(\pi)(\text{length})\left(\left(\frac{width}{2}\right)^2\right)$$

We estimated the number of expected seeds by multiplying the volume by 0.026 (*Fornoni, Valverde & Nunez-Farfan, 2004*). In the laboratory, we opened the fruits to determine the infestation, by counting the weevils. Also, for each fruit we counted the number of remaining, sound, seeds after predation.

## Seed predator's infestation measures

We measured the intensity of infestation by *T. soror* to *D. stramonium* as the number of weevils divided by the number of fruits per plant (*Greischar & Koskella, 2007*). Also, we measured the infestation as the proportion of infested fruits per plant.

## Alkaloid concentration

We quantified the concentration of leaf atropine and scopolamine per plant (*Boros et al., 2010*), by means of high-performance liquid chromatography (HPLC), following *Castillo et al. (2013)*. We obtained the average leaf alkaloid concentration from a sample of 8–10 plants per population. In order to assess if leaf and seeds alkaloid concentration are correlated, we measured alkaloids of mature seeds in a subsample of 119 plants of 14 populations (Table S2).

## Characterization of environmental variables

In order to control for some concomitant environmental variation, the values of mean annual temperature and precipitation for the 31 studied populations of the *D. stramonium* were obtained from Worldclim (*Hijmans & Graham, 2006*). We characterized each population by calculating the Lang's aridity index (*Rehman, 2010*), as the ratio of population

mean annual precipitation and mean annual temperature. Values between 0–20 correspond to deserts, 20–40 arid habitats, 40–60 wet type steppes, 60–100 wet woodlands and 100–160 wet forests (*Perry, 1986*).

### Statistical analyses

We assessed the among-population variation in the intensity of infestation, the proportion of infested fruits by *T. soror*, and the concentration of atropine and scopolamine in seeds and in leaves, using generalized linear model (GzLM), assuming a gamma distribution and a log link function. Our hypothesis is that infestation by the seed predator does not vary among populations of *D. stramonium*.

The relationship between the concentration of atropine and scopolamine in both leaves and seeds was tested through generalized linear mixed models (GzLMM), with a gamma distribution, a log link function and population as a random factor. Data were obtained from individual plants from a subsample of 14 populations. In order to assess the effect of the seed predator on plant fitness, we performed a GzLMM of the number of remaining sound seeds, as estimator of fitness, as a function of the proportion of infested fruits per plant and population as a random factor. Again, we assumed a gamma distribution and a log link function. Similarly, we tested if the concentration of atropine and scopolamine in leaves increases plant individual fitness. We assumed the population × proportion of infested fruits interaction as a random factor.

Selection differentials ($S$) that account for direct and indirect selection acting on a trait were calculated through Pearson correlation coefficient for each population (*Lande & Arnold, 1983*). The concentration of atropine and scopolamine in seeds and leaves was standardized ($x' = \frac{(x-\mu)^2}{\sigma}$). As a proxy of plant fitness we used the estimated number of seeds per plant (other examples in *Kingsolver et al., 2001*). The relative fitness was defined as the ratio between individual fitness and population average fitness. Selection differentials were estimated for each population by correlating alkaloid concentration standardized (atropine or scopolamine) with relative fitness (such in *Sobral et al., 2013*; *Sobral et al., 2015*).

In order to explore the effect of selection exerted by the seed predation and environmental variables on the concentrations of scopolamine and atropine in both leaves and seeds, across populations of *D. stramonium*, we performed generalized linear models (GzLM) (such in *Herrera, Castellanos & Medrano, 2006*; *Sobral et al., 2015*). In each model we included the selection differential of the corresponding alkaloid, latitude, longitude, altitude, and Lang's index of each population. We assumed a gamma distribution (log link function) for alkaloid concentration in leaves, and a Gauss inverse (identity link function) for seeds. We selected the models with lowest corrected AIC values, namely those that explain better the relationship between the variables and consider the sample size (*Akaike, 1974*; *SPSS, 2011*). Statistical analyzes were performed with SPSS v20.0 statistical package (*SPSS, 2011*).

### RESULTS

We detected wide variation in concentration of scopolamine and atropine in leaves (Fig. 2 and Table 1) and seeds (Fig. 3 and Table 1) across populations of *D. stramonium*. The proportion of infested fruits as well as the intensity of infestation by *T. soror* to plants of *D.*

**Table 1** Generalized linear models of the among population variation in the concentration of scopolamine and atropine in leaves and seeds, as well as the intensity of infestation (average number of weevils per fruit, per plant), and the proportion of infested fruits per plant in *Datura stramonium*.

| Response variable | N | d.f. | Wald's chi-square | P | AICc |
|---|---|---|---|---|---|
| Scopolamine of leaves | 278 | 31 | 684.55 | <0.0001 | 345.19 |
| Atropine of leaves | 278 | 31 | 875.11 | <0.0001 | 212.17 |
| Scopolamine of seeds | 119 | 14 | 13441.13 | <0.0001 | 817.18 |
| Atropine of seeds | 119 | 14 | 13062.42 | <0.0001 | 777.75 |
| Intensity of infestation | 859 | 28 | 835.98 | <0.0001 | 74.73 |
| Proportion of infested fruits | 859 | 28 | 1 562.36 | <0.0001 | 1 732.56 |

**Notes.**

AICc, The corrected Akaike information criterion, gives a measure of the relative quality of a statistical model, considering the sample size.

*stramonium* varied significantly among populations (Fig. 4 and Table 1). Variation in the average proportion of infested fruits ranged from zero (populations Coatepec, Huitzuco and Jalapa) up to 90% (populations Teotihuacan and Tlaxiaca; Fig. 4). The average intensity of infestation by *T. soror* varies from populations without infestation up to those with 5 weevils per fruit, per plant.

The concentration of atropine and scopolamine in leaves is positive and significantly related ($N = 117$, $Estimate = 0.289$, $S.E. = 0.055$, $t = 5.285$, $P < 0.0001$, $AICc = 306$), while the correlation of scopolamine in leaves and seeds is negative ($N = 117$, $Estimate = -1.29$, $S.E. = 0.484$, $t = -2.666$, $P = 0.009$, $AICc = 305.279$). Similarly, the concentration of scopolamine in leaves and atropine in seeds are negatively related ($N = 117$, $Estimate = -1.061$, $S.E. = 0.481$, $t = -2.206$, $P = 0.029$, $AICc = 307.407$).

The number of sound remaining seeds per plant showed a negative relationship with the proportion of infested fruits *T. soror* ($N = 278$; $Estimate = -0.629$, $S.E. = 0.266$; $t = -2.367$; $P = 0.019$; $AICc = 832.46$). We found that the number of remaining sound seeds shows a positive relationship with both scopolamine concentration in leaves ($N = 278$; $Estimate = 0.200$, $S.E. = 0.076$; $t = 2.639$; $P = 0.009$; $AICc = 886.85$) and seeds ($N = 119$; $Estimate = 13.56$, $S.E. = 3.56$; $t = 3.81$; $P < 0.0001$; $AICc = 403.32$).

A GzLM of the population average of scopolamine concentration in leaves is positively related to the selection differential ($S$) of leaf scopolamine (Table 2 and Fig. 5A). The same result, although marginally significant, was detected for scopolamine in seeds. In the case atropine concentration in seeds, the GzLM indicates a positive and highly significant relationship with the Lang's aridity index; atropine concentration in leaves is marginally significant in its relationship with Lang's aridity index (Table 2 and Fig. 5B).

## DISCUSSION

Populations of *D. stramonium* vary in the concentration of alkaloids in leaves. This result is in agreement with the study of *Castillo et al. (2013)*. Here, we found that populations of *D. stramonium* also vary in the concentration of alkaloids in seeds, in the intensity of infestation, as well as in the proportion of infested fruits by *T. soror*. The concentration of scopolamine in seeds and leaves is negatively associated across populations. Although such

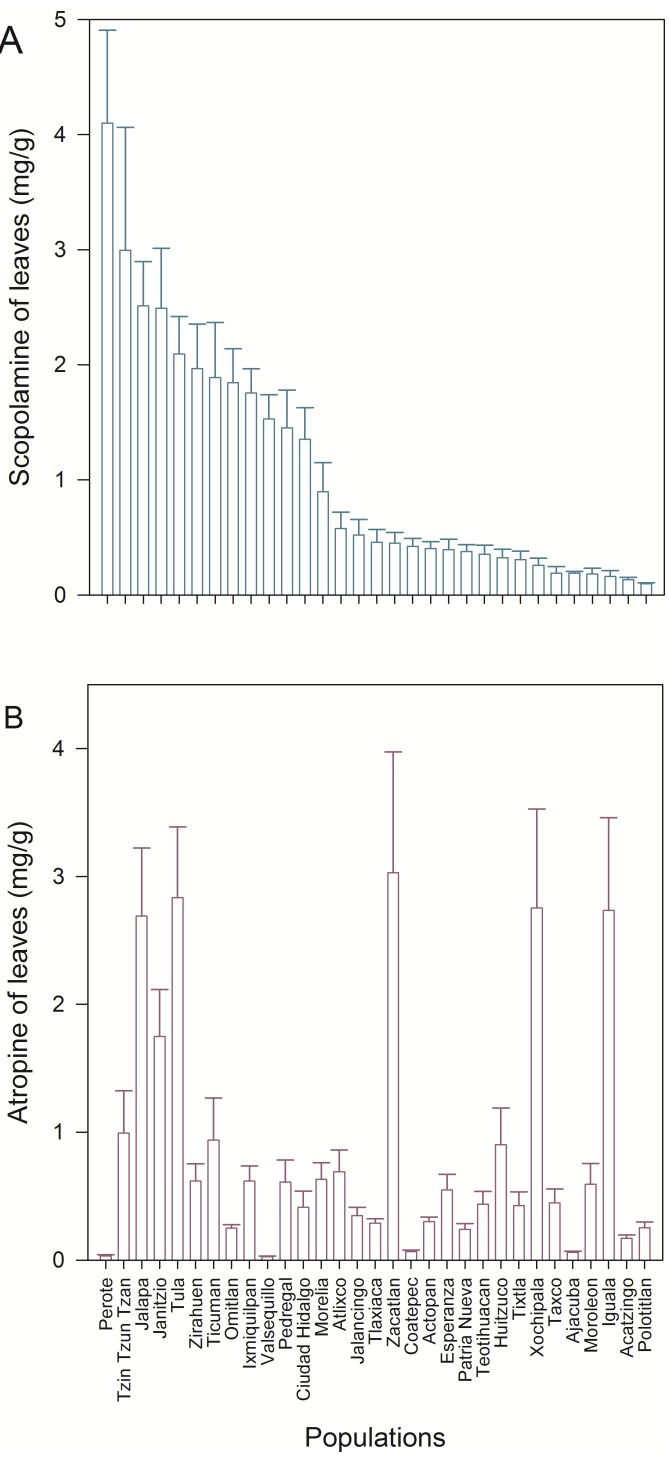

**Figure 2** Average (+1 *S.E.*) concentration of scopolamine (A) and atropine in leaves (B), in 31 populations of *Datura stramonium* from central Mexico.

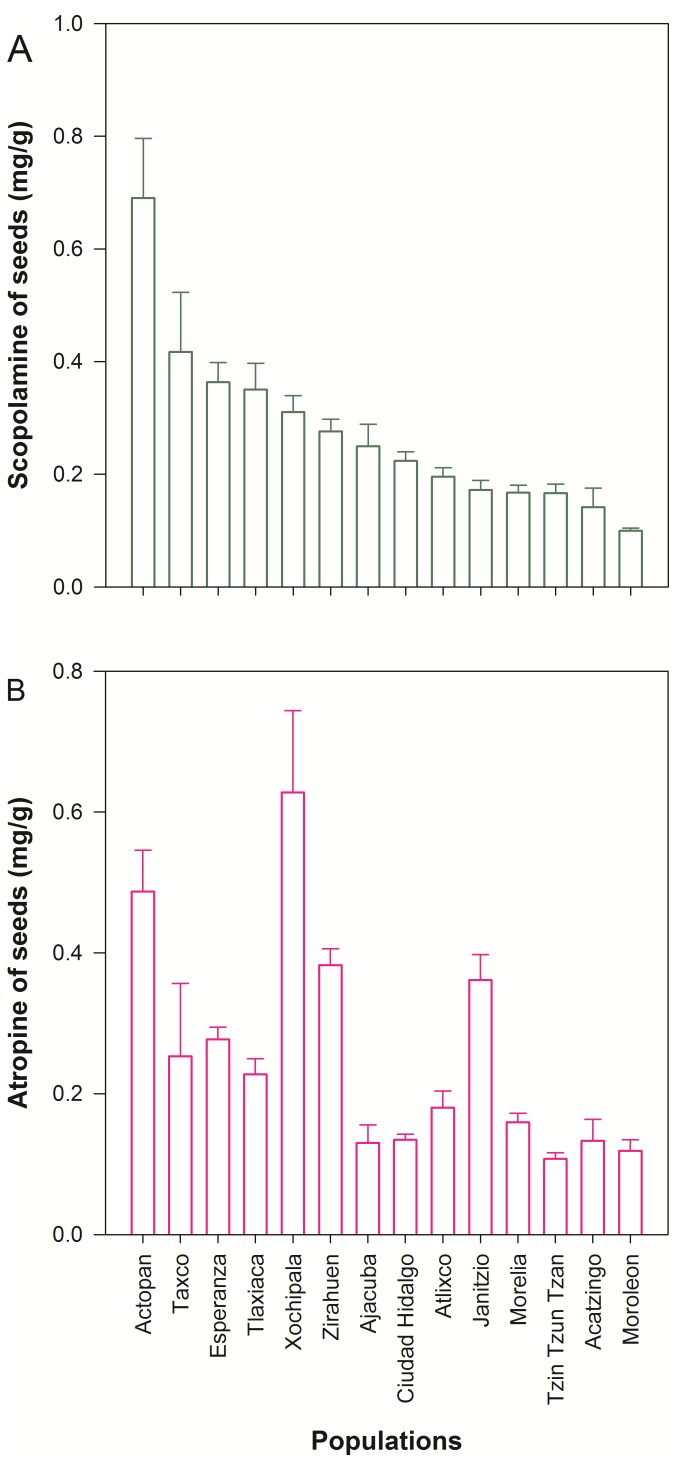

**Figure 3** Average (+1 *S.E.*) concentration of scopolamine (A) and atropine (B) in seeds, in 14 populations of *Datura stramonium* from central Mexico.
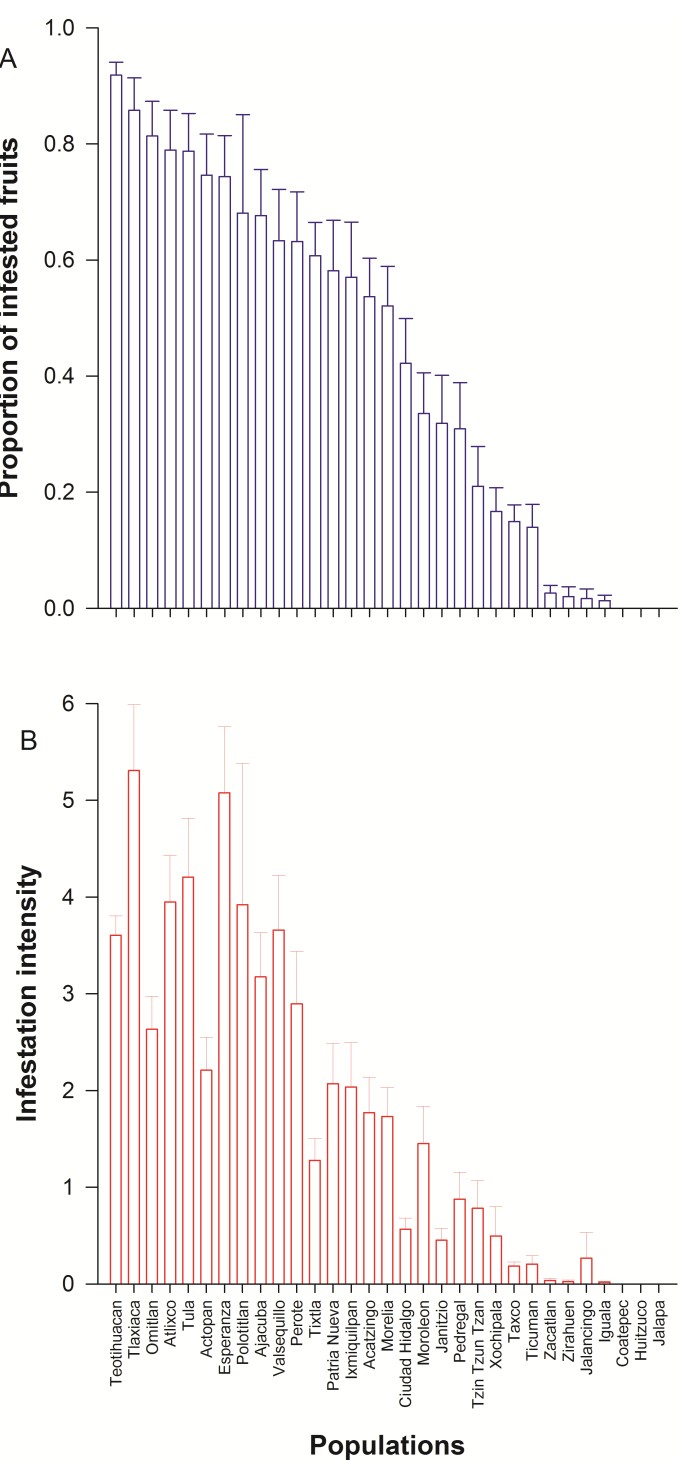

**Figure 4** Average (+1 *S.E.*) proportion of infested fruits per plant (A) and intensity of infestation (the total number of weevils divided by the total number of fruits per plant) (B) by *Trichobaris soror*, in 31 populations of *Datura stramonium* from central Mexico.
**Table 2** Generalized linear models of population average of the concentration of scopolamine and atropine in leaves and seeds of *Datura stramonium*, as a function of the selection differential (S) for the corresponding alkaloid, and environmental variables.

| Response variable | Factors | N | d.f. | Wald's Chi-square | P | AICc |
|---|---|---|---|---|---|---|
| Scopolamine of leaves | Lang's aridity index | 31 | 1 | 0.134 | 0.714 | 72.973 |
| | Altitude | | 1 | 1.563 | 0.211 | |
| | Latitude | | 1 | 0.552 | 0.458 | |
| | Longitude | | 1 | 0.993 | 0.319 | |
| | S Scopolamine | | 1 | 5.662 | 0.017 | |
| Scopolamine of seeds | Lang's aridity index | 14 | 1 | 3.444 | 0.063 | 121.19 |
| | Altitude | | 1 | 2.665 | 0.103 | |
| | Latitude | | 1 | 0.579 | 0.447 | |
| | Longitude | | 1 | 0.912 | 0.339 | |
| | S Scopolamine | | 1 | 3.006 | 0.083 | |
| Atropine of leaves | Lang's aridity index | 31 | 1 | 3.557 | 0.059 | 61.072 |
| | Altitude | | 1 | 3.419 | 0.064 | |
| | Latitude | | 1 | 0.004 | 0.95 | |
| | Longitude | | 1 | 0.001 | 0.976 | |
| | S Atropine | | 1 | 0.812 | 0.367 | |
| Atropine of seeds | Lang's aridity index | 14 | 1 | 11.291 | 0.001 | 122.06 |
| | Altitude | | 1 | 0.002 | 0.967 | |
| | Latitude | | 1 | 1.666 | 0.197 | |
| | Longitude | | 1 | 1.288 | 0.256 | |
| | S Atropine | | 1 | 0.011 | 0.918 | |

**Notes.**

AICc, The corrected Akaike information criterion, gives a measure of the relative quality of a statistical model, considering the sample size.

a pattern was not detected for atropine, the concentration of scopolamine and atropine in leaves, positively covary across populations. Remarkably, we detected that scopolamine concentration in both leaves and seeds enhances individual plant fitness. Our major finding was the detection of a positive relationship between the population average concentration of scopolamine in both leaves and seeds with the selection differentials of scopolamine. This implies that natural selection explains the among population variation in scopolamine concentration. Thus, *T. soror* is driving, at least in part, the variation in chemical defense in *D. stramonium* (*Castillo et al., 2015*).

A previous study has reported that scopolamine plays a role in the interaction between *D. stramonium* and its main folivore insects in central Mexico (*Castillo et al., 2014*). Here, we found that among-populations of *D. stramonium*, plants with higher concentration of scopolamine in leaves had a higher number of remaining sound seeds. This suggests that scopolamine acts as a defense character against *T. soror*, resulting in fewer consumed or damaged seeds. A similar trend has been found in the hemiparasitic plant *Castilleja indivisa* (*Adler, 2000*), where the alkaloid lupanine, obtained from its host plant, *Lupinus texensis*, reduces damage to its flowers by herbivores and increases visitation by pollinators, thus enhancing plant fitness, measured as the number of seeds.

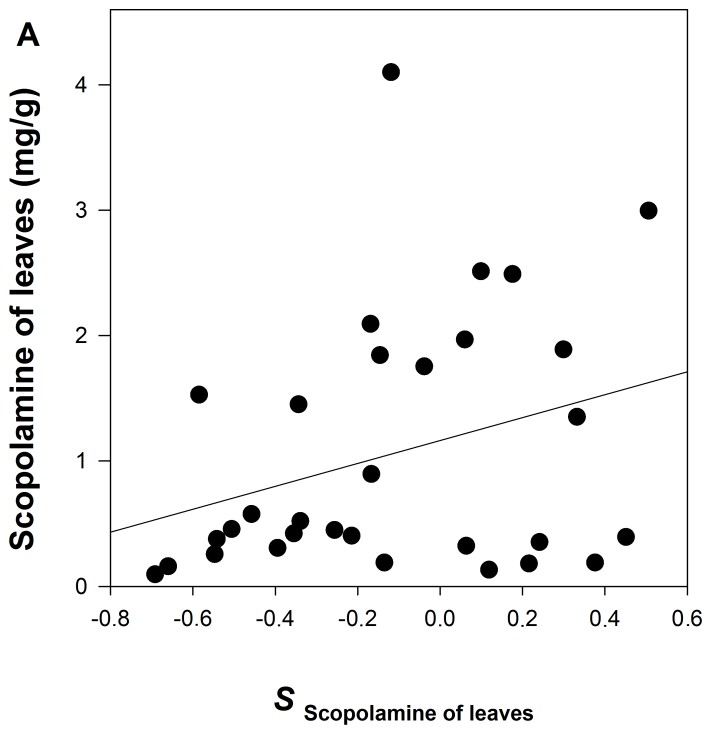

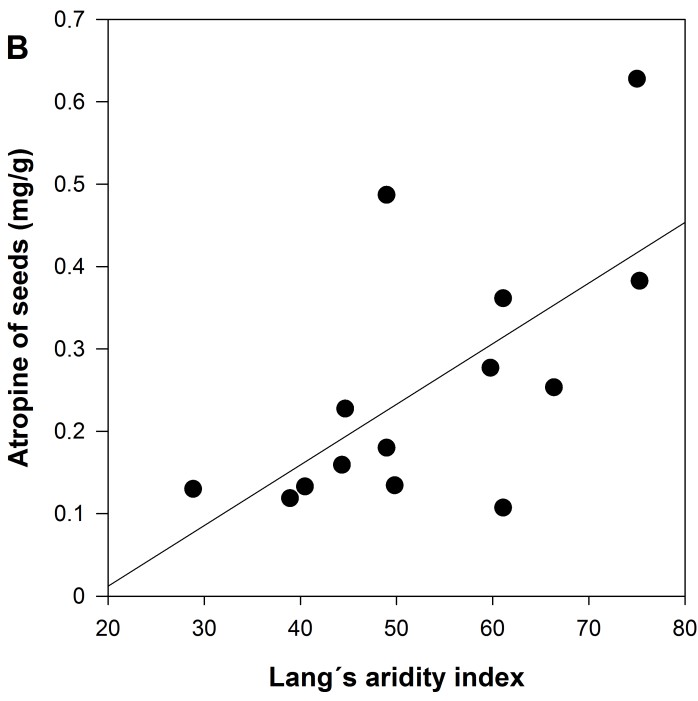

**Figure 5** Relationships between (A) the population average of scopolamine concentration in leaves with $S_{\text{Scopolamine of leaves}}$ ($\rho = 0.3079$), and (B) the population average of atropine concentrations in seeds with the Lang's index ($\rho = 0.6434$).

The fact that the concentration of scopolamine in leaves and seeds are negatively correlated, probably pointing an allocation trade-off (*Kariño-Betancourt et al., 2015*), this does not preclude that scopolamine concentration in leaves had a negative effect on the seed predator. Adult females and males of *T. soror* courtship, mate, and feed on *D. stramonium* leaves (J Núñez-Farfán, pers. obs., 1989). When feeding, adult weevils produce small holes on the leaf blade, and sometimes damage can be severe (*Cabrales-Vargas, 1991*). Thus, it is likely that *T. soror*'s females, while feeding, might "assess" the chemical resistance level of a plant (i.e., atropine/scopolamine concentration in leaves) and select which plants are suitable to oviposit. This would result in lower or null oviposition in those plants with high concentration of scopolamine in leaves. Some evidence in this line shows a close relationship between *Trichobaris* and *Datura*. For instance, *Trichobaris bridwelli* oviposits preferentially on *D. stramonium* rather than on tobacco plants (*Nicotiana tabacum*) (*Cuda & Burke, 1991*), since *T. bridwelli* does not tolerate the pyridine alkaloids of tobacco. On the other hand, *Diezel, Kessler & Baldwin (2011)* have experimentally demonstrated that *T. mucorea*, a species that burrows into the stems of *Nicotiana attenuata*, increases infestation on transgenic plants of *N. attenuata* where the chemical defenses (jasmonic acid and nicotine) were silenced.

The role of scopolamine as defense against herbivory in *D. stramonium* is supported by the findings of *Castillo et al. (2014)*. In such study, they detected positive selection on scopolamine concentration in leaves in two populations of *D. stramonium,* one preyed upon by the dietary specialist *Lema daturaphila*, and the other by the dietary generalist *Sphenarium purpurascens*. However, negative selection on scopolamine was also detected in a third population of *D. stramonium* preyed upon by *L. daturaphila*. Thus, the resistance role of tropane alkaloids of *D. stramonium* varies according to the type of herbivore and the tissue they consume, as well as the environmental conditions of populations (*Castillo et al., 2014*). Further studies are needed to determine the independent and join selective effects of folivores and seed predators on the concentration of scopolamine in seeds and leaves of *D. stramonium*.

The GzLM shows that atropine in seeds covaries positively with the Lang's index. That is to say, populations of dry environments have less atropine concentration. It has been reported that alkaloid production may be water limited, decreasing in concentration in dry environments (*Baricevic et al., 1999*). In the interaction between *Curculio camelliae-Camellia japonica*, the expression of defense is also affected by the habitat's environmental conditions; infestation by the weevil was higher at high altitudes, where the fruits of *C. japonica* trees have thinner pericarps (*Toju, 2009*). Thus, the evolution of chemical defense against herbivores in *D. stramonium* can also be environmentally conditioned.

We found evidence of natural selection on chemical resistance of *D. stramonium* since higher scopolamine concentration increases plant fitness. The GzLM analysis revealed a positive relationship between the population average concentrations of scopolamine in both leaves and seeds with their selection differential of scopolamine. This relationship suggests that the among-population variation in chemical defense of *D. stramonium* is molded by *T. soror*. Thus, populations of *D. stramonium* vary in the direction and strength of selection on chemical defense, an expectation of the geographic mosaic of coevolution

(i.e., hot spots and cold spots; *Thompson, 2005*). Previous evidence has demonstrated that leaf herbivores exert selection pressures over physical and chemical defenses of *D. stramonium* (*Valverde, Fornoni & Núñez-Farfán, 2001*; *Castillo et al., 2014*). This is the first evidence that seed predators also exert a significant selective pressure on chemical defense of *D. stramonium*.

## ACKNOWLEDGEMENTS

We thank to Luis Barbo for helping us during HPLC quantification, and the members of Laboratorio de Genética Ecológica y Evolución for logistic support and field assistance. We are grateful to the reviewers and Dr. Mar Sobral of helpful statistical advice. This paper constitutes a partial fulfillment of A. M-P s Doctoral dissertation in Programa de Doctorado en Ciencias Biomédicas, Universidad Nacional Autónoma de México (UNAM).

### Funding

A. M-P acknowledges the scholarship for graduate studies (251620) granted by the National Council of Science and Technology (CONACyT). This project was funded by a CONACyT grant to J. N-F. The funders had no role in study design, data collection and analysis, decision to publish, or preparation of the manuscript.

### Grant Disclosures

The following grant information was disclosed by the authors:
National Council of Science and Technology (CONACyT): 251620.

### Competing Interests

The authors declare there are no competing interests.

### Author Contributions

- Adán Miranda-Pérez conceived and designed the experiments, performed the experiments, analyzed the data, wrote the paper, prepared figures and/or tables, reviewed drafts of the paper.
- Guillermo Castillo and Johnattan Hernández-Cumplido performed the experiments, reviewed drafts of the paper.
- Pedro L. Valverde conceived and designed the experiments, performed the experiments, analyzed the data, wrote the paper, reviewed drafts of the paper.
- María Borbolla, Laura L. Cruz and Rosalinda Tapia-López performed the experiments.
- Juan Fornoni conceived and designed the experiments, performed the experiments, reviewed drafts of the paper.
- César M. Flores-Ortiz performed the experiments, contributed reagents/materials/analysis tools.
- Juan Núñez-Farfán conceived and designed the experiments, performed the experiments, analyzed the data, contributed reagents/materials/analysis tools, wrote the paper, reviewed drafts of the paper.

## Data Availability

The raw data has been supplied as Data S1.

## Supplemental Information

Supplemental information for this article can be found online at http://dx.doi.org/10.7717/peerj.1898#supplemental-information.

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
