# Peer review of "Natural selection drives chemical resistance of Datura stramonium"

_PeerJ, doi:10.7717/peerj.1898_

## Round 0.1 · original submission · Minor Revisions

Both reviewers agree that the study is valuable and performed well. Both have some minor comments regarding mainly formal things, which should be easy to comply. Please, also take in consideration the attached annotated manuscript.

In addition to this reviewer 2 has suggestions about alternative or additional forms of statistical analysis. I let it for the decision of the authors if they want to pick up these suggestions. If they do not want to change their analysis, please give some comments about this decision.

Reviewer 1 ·

Basic reporting

This is an excellent study. The questions are clear and the analyses are appropriate.

The Tables and Figures need some clarifications to help readers.

Table 1: add sample sizes

Table 2: In the table caption, identify T as the test statistical and expand the acronym AIC and and briefly explain how it is used here. Most readers will know, but everything in a table should be briefly identified in table captions.

Figure 2: Briefly explain in the table caption the meaning of infestation intensity so that readers can understand the scale on the Y-axis without going back to the Methods.

Figures 3 and 4: Give correlation values and p-values on the figures so that readers can assess the lines that are shown

Figure 4: Indicate whether the error bars are standard errors, confidence limits, or something else.

The manuscript should be proof-read for a few remaining typographical errors (e.g., unknow rather than unknown in one place; a missing "to" on line 289).

I do not know if "All appropriate raw data have been made available..."

Experimental design

The methods are clearly described and the statistical analyses are appropriate. There are multiple ways of analyzing results such as these, but the authors have taken a reasonable approach.

Validity of the findings

The results are convincing and suggest strong, geographically variable, and potentially reciprocal, selection on these interacting species. This study appears to have been carefully done, carefully interpreted, and not over-interpreted. The conclusions do not go beyond the results. This is a very good paper, requiring considerable field and laboratory work. It is among the few studies have have assessed selection differentials among potentially coevolving species in many populations. The results suggest ongoing selection on the species and potentially ongoing reciprocal selection in some localities.

Additional comments

This is a nicely designed and analyzed study, suggesting the potential of a geographic mosaic of coevolution between these species.

·

Basic reporting

No Comments

Experimental design

This is a great dataset with a correct “experimental” design. With those data it is possible to study how T. soror affects defensive alkaloids in seeds and leaves of the plants and whether these selective pressures affect the geographic variation on plant traits –such as the variation on levels of defensive compounds.

However the current treatment of data is not completely satisfactory. Important but easy-to-fix changes in the analytical approach should we made in order to make this research correct.

The general approach is great but there are some mistakes regarding the types of model used, or even the response variable in the latest two models (which are the more important in this ms). Note that I took the opportunity to analyze the data myself (from the dataset in the submission) and actually found that the effect of natural selection drives among populations variation on the concentration of alkaloids, thus the main message in the manuscript is not going to change once it is properly analyzed.

I think that the phenotypic trait which should be studied in the latest two models is the content on defensive compounds. It should be analyzed how quantity of atropine and scopolamine in seeds and leaves (or altogether) affect the number of seeds surviving from predation IN EACH PLANT. It would be also possible to make combinations of these four traits (for example, total alkaloids in seeds or in leafs, or total atropine or scopolamine in plants).
The first step (which in my opinion is optional) could be to analyze how number of seeds saved from predation depends on the population, the number of fruits per plant, the atropine in seeds, atropine in leafs, scopolamine in leafs, scopolamine in seeds (correlations between phenotypic traits should be taken into controlling for colinearity issues). In this model interaction between population and each of the phenotypic traits should be included. This would shed some light in the issue of whether defensive compounds affect the reproductive output of plants (after predation, i.e. the number of seeds saved from predation) and whether that relationship is constant or varies among populations.
Second step would be what you already have correctly performed. You properly run phenotypic selection analyses at the level of population. By regressing each phenotypic trait (for example atropine in seeds) on the plant relative fitness (number of seed saved from predation relative to the average of the population) selective differentials are assessed (S). Also you understood, and I agree, that the important question in here is not the significance of those selective differentials within populations. The relevant question here is addressed on the next step –which is studying how the selective pressures explain variation on phenotypic traits among populations.
On the third step selective pressures should be covariates explaining phenotypic traits at the population level (one model for each trait as response variable, one for average population atropine in seeds, another for average population scopolamine in seeds etc.). This is where you performed an important conceptual mistake by running the models for the response variable “infestation” instead of the selected phenotypic traits which would be “alkaloids”. I agree with you with the importance of adding environmental covariates to control for other sources of variation in these models.
If selective differentials are found to be significant in explaining the among population variation on defensive traits (and my analyses using your dataset shows they are indeed), then it would be actually possible to state that “natural selection by T. soror drives, at least in part, the variation on defensive compounds across D. stramonium populations”.

Some minor considerations (Please also see notes on attached pdf)
• Please use GLM instead of LMM because your response variables are not normal. It is better to analyze actual distribution of variables by means of Generalized Linear Models which allow fitting response variables to different error distributions and link functions depending on the nature of the response variables.
• Please send table 1 to supplementary materials.
• Analyses in figure 3 should be done at the plant level including populations as random factor (not using population averages) and only necessary for the sound seeds, the rest (in my opinion) is redundant. You are trying to show that infestation changes fitness. If you use sound seeds as response, and infestation levels as predictors, at the same time population as random factor that would be a better approach. If you add the interaction of population with infestation level, and it proves significant, that would show how selective pressures by weevils on defenses vary among population. Another possibility is to escape this step altogether and show the analyses done only from the calculations of selection coefficients on.
• In the dataset and in the manuscript it is not clear when showing results on alkaloids if results are on alkaloids of seeds or leaves or a combination of both. In the next version of the paper this information should be clarified.

Validity of the findings

No comments

Additional comments

The manuscript “Natural selection drives chemical resistance of Datura stramonium to its specialized seed predator” constitutes a very interesting piece of information. The study of effects of natural selection on geographic variation of plant traits is not commonly addressed. This is a very meritorious approach which is very scarce in literature.

The research aimed to find out if seed predation by T. soror on Datura stramonium caused selective pressures and if this natural selection drives differences in plant resistance among populations. It was already known that atropine in this species is a defensive compound to leaf herbivory but it was not known whether this defensive compound was also related to predation upon seeds.

Besides this interesting approach I find another major strength on this manuscript. I commend the authors on the nice dataset they have collected. Ecological studies comprising a large number of populations across broad geographical scales are uncommon –despite many evolutionary questions in ecology should be addressed from this perspective. This study comprises 31 populations of D. stramonium across central Mexico in which between 21 and 30 individuals per population have been studied, although actually between 7 and 10 plants per populations were used in the main analyses. The level of infestation by T. soror and the levels of defensive alkaloids (atropine and scopolamine) both in leaves and seeds for 7-10 plants per population (278 plants) have been measured. Additionally, some environmental covariates at the level of populations (mean annual temperature and precipitation) have been collected –which is a very good idea so that some environmental variation can be controlled in analyses showing how T.soror affects defensive compounds in D. stramonium.

---

## Round 0.2 · Minor Revisions

Both reviewers agree that the manuscript was significantly improved.

There are remaining minor concerns about typos in the manuscript. Please, review your manuscript carefully for typing errors, before it can finally be accepted.

Reviewer 1 ·

Basic reporting

Very good. The initial version was good and this version is even better.

Experimental design

Good. I can see the argument for changing the statistical analyses and think that the authors did a good job on those reanalyses.

Validity of the findings

The results seem to be carefully interpreted.

Additional comments

The manuscript could use one more reading for typos, for example:
line 234: length, not lenght
line 578: This points that..." A word seems to be missing
line: 705: predator, add s

·

Basic reporting

No Comments

Experimental design

No Comments

Validity of the findings

No Comments

Additional comments

The manuscript is greatly improved.

In my opinion this article is very meritorius, clear and properly executed. Congrats for the nice work

---

## Round 0.3 · accepted · Accept

Congratulations to a very good manuscript.